# An Embedded Mixed-Methods Study with a Dominant Quantitative Strand: The Knowledge of Jordanian Mothers About Risk Factors for Childhood Hearing Loss

**DOI:** 10.3390/audiolres15040087

**Published:** 2025-07-16

**Authors:** Shawkat Altamimi, Mohamed Tawalbeh, Omar Shawkat Al Tamimi, Tariq N. Al-Shatanawi, Saba’ Azzam Jarrar, Eftekhar Khalid Al Zoubi, Aya Shawkat Altamimi, Ensaf Almomani

**Affiliations:** 1Department of Special Surgery, Faculty of Medicine, Al-Balqa Applied University, Al-Salt 19117, Jordan; saba.jarrar@bau.edu.jo; 2Department of Special Surgery, School of Medicine, University of Jordan, Amman 11942, Jordan; tawalbeh@ju.edu.jo; 3King Hussein Medical Center, Amman 11733, Jordan; prof.tamimi@hotmail.com; 4Department of Public Health and Community Medicine, Faculty of Medicine, Al-Balqa Applied University, Al-Salt 19117, Jordan; talshatanawi@bau.edu.jo; 5New Al-Hussein Hospital, Al-Salt 19117, Jordan; zoubi24972@hotmail.com; 6Department of Basic Medical Science, Faculty of Medicine, Al-Balqa Applied University, Al-Salt 19117, Jordan; aya.tamimi@bau.edu.jo (A.S.A.); ensaf.momani@bau.edu.jo (E.A.)

**Keywords:** childhood hearing loss, maternal knowledge, health literacy, socioeconomic status, early detection, Jordan

## Abstract

**Background**: Childhood hearing loss is a public health problem of critical importance associated with speech development, academic achievement, and quality of life. Parents’ awareness and knowledge about risk factors contribute to early detection and timely intervention.  **Objective**: This study aims to examine Jordanian mothers’ knowledge of childhood hearing loss risk factors and investigate the impact of education level and socioeconomic status (SES) on the accuracy and comprehensiveness of this knowledge with the moderating effect of health literacy. **Material and Methods**: The approach employed an embedded mixed-methods design with a dominant quantitative strand supported by qualitative data, utilizing quantitative surveys (n = 250), analyzed using structural equation modeling (SEM) in SmartPLS, and qualitative interviews (n = 10), analyzed thematically to expand upon the quantitative findings by exploring barriers to awareness and healthcare-seeking behaviors. **Results**: The accuracy and comprehensiveness of knowledge of hearing loss risk factors were also positively influenced by maternal knowledge of hearing loss risk factors. Maternal knowledge was significantly associated with both education level and socioeconomic status (SES). Furthermore, maternal knowledge and accuracy were significantly moderated by health literacy, such that mothers with higher health literacy exhibited a stronger relationship between knowledge and accuracy. Qualitative findings revealed that individuals encountered barriers to accessing reliable information and comprehending medical advice and faced financial difficulties due to limited options for healthcare services. **Conclusions**: These results underscore the need for maternal education programs that address specific issues, provide simplified healthcare communication, and enhance access to pediatric audiology services. Future research should explore longitudinal assessments and intervention-based strategies to enhance mothers’ awareness and detect early childhood hearing loss.

## 1. Introduction

Childhood hearing loss is a significant public health challenge that affects language acquisition, cognitive development, academic achievement, and socialization [1]. There are more than 34 million children across the world with disabling hearing loss, mostly in low- and middle-income countries where hearing loss among children remains undiagnosed due to limited early screening and treatment services [2]. Hearing loss in childhood remains unidentified because of parent ignorance, cultural misunderstanding, and the inaccessibility of treatment facilities. The long-term impact of hearing loss can be mitigated through early diagnosis and intervention, with the involvement of parents, especially mothers [3].

Adequate knowledge among mothers regarding risk factors, signs, and warning signs of hearing loss, as well as accessible healthcare services, is fundamental for early detection and timely care [4]. Childhood hearing loss is associated with preventable risk factors, including ototoxic medications, prenatal infections, noise exposure, premature births, and genetic predisposition, which are essential modifiable determinants of childhood hearing loss [5,6,7]. Nevertheless, individuals from various cultural, educational, and socioeconomic backgrounds differ substantially in their awareness of these components, which influences the ability to diagnose early and take measures [8]. There can be several reasons behind children’s hearing loss, which can broadly be grouped into genetic, prenatal, perinatal, postnatal, and environmental/lifestyle factors. Figure 1 shows the principal categories of such risk factors.

In most developing nations, such as those in the Middle East and North Africa (MENA), inconsistent knowledge in mothers puts children at a higher likelihood of delayed diagnosis and developmental effects [9]. Most MENA countries, including Jordan, present sociocultural and economic challenges peculiar to the global situation of childhood hearing loss. Although the field has advanced in terms of the development of hearing screening and healthcare delivery platforms in general, Jordan has also failed to properly sensitize mothers concerning risk factors and the ongoing detection of hearing loss [10]. The understanding of risk factors for hearing loss and the importance of detecting it early varies among mothers, depending on their education, socioeconomic status (SES), and health literacy. Consanguineous marriages occur at a high rate in Jordan and increase the chances of genetic hearing disorder; thus, it is important to educate mothers on these risks [11].

This paper discusses parental awareness of the risk factors for childhood hearing loss, with an emphasis on maternal education, socioeconomic status (SES), and health literacy. In this regard, this research will adopt a mixed-methods approach to yield both quantitative knowledge levels and qualitative insights into maternal awareness, healthcare-seeking behavior, and barriers to healthcare services. To design effective maternal health education programs and particular interventions in the early detection and prevention of childhood hearing loss in Jordan, several factors need to be understood [12].

Hearing loss in children is a significant global public health issue affecting speech, language development, academic performance, and quality of life [1]. Early detection and intervention can minimize adverse effects; however, parental awareness and knowledge hinder their success [5]. This research establishes maternal understanding of the early indications of hearing loss and the need for early healthcare steps [8,12]. Despite this, there is very low knowledge of hearing loss risk factors in these countries, particularly in developing countries and in the Middle East [3,7].

Thus, there is little research on this issue in Jordan, where cultural, economic, and healthcare accessibility factors influence maternal knowledge. Studies from neighboring Arab countries, such as Egypt and Saudi Arabia, support the notion that parental knowledge is primarily based on parents’ educational level, socioeconomic status (SES), or lack of access to healthcare [4,8]. However, little empirical evidence has been obtained regarding how these factors affect Jordanian mothers’ knowledge and perception of hearing loss risk factors. Thus, maternal knowledge information is important in this context as Jordan has a high rate of consanguineous marriage (a genetic risk for congenital hearing loss) [12]. Gaps in maternal awareness may result in delayed diagnosis or in early intervention being unavailable, leading to fewer educational and social opportunities for children affected by it [2,9]. It has also been suggested that their limited health literacy may delay healthcare-seeking behavior, which in turn impedes the long-term management of hearing loss [13].

Despite the research on parental awareness of childhood deafness carried out in other countries, little research has been conducted regarding Jordanian mothers [7,9]. This results in an empirical gap in understanding how maternal education, socioeconomic status (SES), and health literacy relate to awareness of hearing loss risk factors. In most existing studies, however, descriptive assessments are provided without considering health behavior theories related to knowledge acquisition and healthcare decision-making [3,13]. There are several models, such as the Health Belief Model (HBM) and the Theory of Planned Behavior (TPB), that could provide much deeper insights into why some mothers are more knowledgeable than others.

Previous research relied on useful quantitative cross-sectional studies, which could not reveal why maternal awareness changes and how this happens [5,8]. A mixed-methods, quantitative and qualitative approach is needed to explore both statistical associations and contextual factors that shape maternal knowledge.

Regardless of regional disease awareness campaigns aimed at implementing early screening and parental education, several mothers still lack awareness of critical risk factors, including ototoxic medications, pregnancy infections, and genetic predisposition [14]. In Jordan, there are no targeted awareness programs, and no studies have been conducted to determine whether mothers’ health literacy affects their ability to understand and utilize health information. It would be interesting to learn from the insights they provide to policymakers, healthcare providers, and educators, allowing them to address these gaps in terms of empirical, theoretical, methodological, and practical aspects. The findings will inform the development of culturally appropriate interventions to enhance maternal awareness, increase early screening uptake, and promote child health outcomes in Jordan [11].

Although there have been attempts globally to enhance the early detection of hearing loss in children, the issue of maternal knowledge in early identification and intervention in Jordan has not been explored yet. Several researchers have studied parental awareness in different Middle Eastern and African settings [4,5,8], but none systematically looked at how maternal education, financial status (socioeconomic status), and literacy regarding health affect awareness of the risk factors of hearing loss in Jordan. Considering the social and cultural context of the country, including the high frequency of consanguineous marriage, a known genetic risk factor for its most prominent genetic disease, this study contributes empirically to knowledge of an understudied population. By examining these maternal factors, the study aims to determine whether mothers can facilitate early detection and intervention for hearing loss.

This study brings more than an empirical contribution; it also presents a methodological advancement using mixed methodology (qualitative interviews and quantitative surveys). Most previous studies on maternal awareness were based on cross-sectional surveys, which cannot fully portray in-depth perceptions and contextual influences on maternal knowledge. Through the combination of statistical analysis and the deliberation of rich qualitative insights, this study offers a holistic account of the barriers and factors likely to influence mothers’ knowledge [3]. In addition, it enriches theoretical discussions in the field by integrating models of health behavior such as the Health Belief Model (HBM) and the Theory of Planned Behavior (TPB) to understand how maternal perception, belief, and literacy levels affect health-seeking behavior. This study’s contribution is enhanced by the application of these theoretical frameworks, thereby bridging the gap between knowledge acquisition and real-world health decisions.

### 1.1. Accuracy and Comprehensiveness of Knowledge and Maternal Knowledge of Hearing Loss Risk Factors

The knowledge of mothers is vital in the early detection and prevention of childhood hearing loss, particularly in terms of its impact on parental decision-making and health-seeking behaviors [12]. Mothers who are aware of the risk factors for hearing loss are more likely to identify early warning signs and seek medical help when needed [5]. Risk factors include genetic predisposition, prenatal infections (such as rubella and cytomegalovirus), perinatal complications (such as preterm birth and neonatal hyperbilirubinemia), and exposure to ototoxic medications, as well as environmental factors (such as noise exposure) [7]. Nevertheless, maternal misconceptions and the absence of trustworthy health info prevent early diagnosis and lead to inadequate management of children’s hearing loss [15]. While studies have demonstrated that better maternal knowledge about health is related to better child health, enormous variation in the accuracy and coverage of maternal knowledge also exists [9]. In settings where resources are scarce, mothers tend to have little awareness of health education programs and then rely on outdated beliefs or misinformation, which may hinder their ability to adopt proactive health measures [5]. 

### 1.2. The Role of Mothers’ Education Level in Maternal Knowledge of Hearing Loss Risk Factors

Education is considered a significant factor in determining health awareness and healthcare practices [13]. Research suggests that the higher a mother’s education level is, the more likely she is to have accurate knowledge about childhood diseases [8]. Additionally, educated mothers are more capable of understanding medical information, recognizing symptoms, and determining whether intervention is needed earlier in seeking treatment [2]. A study in South Africa demonstrated that mothers with tertiary education were more aware of the risk factors for hearing loss than mothers with lower educational levels [3]. For instance, women with less-educated backgrounds may not be able to access scientifically right information related to their health and may put more trust in traditional beliefs or informal sources of information that lead to misconceptions about the prevention of hearing loss [7]. 

### 1.3. The Impact of Socioeconomic Status (SES) on Maternal Knowledge of Hearing Loss Risk Factors

Socioeconomic status (SES) matters in terms of educational opportunities and access to healthcare and health information [5], and so it plays a part in determining the level of health knowledge. Mothers with higher-socioeconomic-status (SES) backgrounds tend to have greater access to healthcare facilities and preventive health programs, as well as numerous awareness campaigns that enable them to be aware of risk factors and early intervention for hearing loss [9]. Mothers from higher income levels are better at identifying the early signs of hearing loss and seeking medical attention earlier than those from lower income levels [7]. On the other hand, mothers from low-SES backgrounds have barriers of access to healthcare such as financial capacity, low exposure to health education programs, and low primary healthcare-seeking behaviors [3]. Traditionally, low-income mothers tend to rely more on traditional beliefs or knowledge from their community, which may or may not be scientifically correct [8]. Consequently, more maternal knowledge of hearing loss risk should be observed to be associated with higher SES.

### 1.4. Moderating Role of Health Literacy

Health literacy refers to the knowledge encompassing one’s ability to comprehend, decode, process, and employ health-related information in health decision-making [6]. Mothers with health literacy are able to understand health-related information and translate or critically analyze it. Additionally, their proper interaction with healthcare professionals provides more accurate and broader knowledge [9]. For example, a Pakistani study found that mothers with better health literacy are more comfortable identifying early signs of childhood hearing loss and asking for professional assistance than those with less health literacy [13]. However, on the other hand, low health literacy levels may lead to errors in interpreting medical advice, reliance on incorrect health beliefs, and delayed healthcare-seeking behaviors [15]. There are gaps in mothers’ knowledge, despite exposure to health education, because they are likely to have low literacy levels and struggle to understand and apply medical terms correctly [8]. 

### 1.5. Objective

The findings of this study have practical significance for healthcare policymakers, educators, and public health professionals in Jordan and other low- and middle-income countries facing similar challenges. The study identifies key determinants of maternal knowledge. It provides evidence-based recommendations for designing health awareness campaigns and community-based interventions to promote the early detection of childhood hearing loss. Additionally, understanding how health literacy affects maternal decision-making will enable healthcare providers to employ more effective communication strategies to disseminate and utilize critical health information [10].

This study aims to examine Jordanian mothers’ knowledge of childhood hearing loss risk factors and investigate the impact of education level and socioeconomic status (SES) on the accuracy and comprehensiveness of this knowledge with the moderating effect of health literacy. The hypotheses of this study are as follows:

**H1.** *Maternal knowledge of hearing loss risk factors plays a significant role in the accuracy and comprehensiveness of maternal knowledge*.

**H2.** *Maternal awareness of risk factors for hearing loss is positively associated with mothers’ education levels*.

**H3.** *There is a positive association between socioeconomic status (SES) and maternal knowledge about hearing loss risk factors*.

**H4.** *The relationship between maternal knowledge and the accuracy of maternal knowledge is moderated by maternal health literacy level*.

## 2. Materials and Methods

### 2.1. Research Design

An embedded mixed-methods approach with a dominant quantitative strand was adopted in this study. The quantitative phase consisted of a cross-sectional survey assessing maternal knowledge levels, whereas the qualitative phase involved semi-structured interviews on perceptions, experiences, and barriers to awareness. The qualitative dataset was applied to complement and contextualize the results of the quantitative analysis, identifying more widely how factors may have been affecting maternal knowledge and healthcare-seeking behavior. Data triangulation was achieved through a mixed-methods approach, ensuring both statistical validity and contextual depth [16]. In quantitative analysis, the focus was the relationship between maternal knowledge, the accuracy of knowledge, socioeconomic status (SES), education level, and health literacy. The qualitative analysis provided a deeper understanding of how and why these factors influence maternal awareness and the determinants of healthcare-seeking behavior [3].

### 2.2. Population and Sampling

The target population was Jordanian mothers whose children were aged between 0 and 10 years, as this is an important age range in which to detect early signs of hearing loss. The recruitment process was facilitated by the assistance of hospitals, maternal health clinics, audiology clinics, and online parenting communities. To qualify for inclusion, participants had to have a child in the targeted age range. Mothers whose children had been diagnosed with severe or profound hearing loss before the study or mothers who did not communicate in Arabic were excluded. This provided an appropriate representation of mothers whose children were at risk of developing hearing loss. Stratified sampling was employed during the quantitative stage to represent different levels of education and income. The 250 mothers who were selected as a sample were stratified according to their educational level (no formal education, primary, secondary, university) and their income level (low, medium, high). Individuals were randomly chosen in each stratum to ensure maximum representation of samples with diverse socioeconomic backgrounds. Structural equation modeling (SEM) was appropriate for the sample size since it had sufficient statistical power to detect such a small relationship. Purposive sampling was adopted in the qualitative phase to include 10 mothers with diverse educational levels, socioeconomic backgrounds, and health literacy levels. The sample was evaluated based on data saturation, which means that no new themes were identified after a certain number of interviewees had been interviewed. The qualitative findings were then used to elaborate on and contextualize the quantitative study’s findings, providing a better understanding of the barriers and influencing factors that impacted maternal knowledge and healthcare-seeking behavior. The qualitative phase was expected to provide interpretations of and substantiate the associations identified in the quantitative study, which would aid in determining the reasons why some factors led to maternal knowledge of childhood hearing loss.

### 2.3. Data Collection Methods

Quantitative and qualitative data were collected for this study to assess Jordanian mothers’ knowledge of the risk factors for childhood hearing loss. The data were collected through a structured questionnaire and semi-structured interviews (see the Appendix A).

For the quantitative phase, a self-administered questionnaire was distributed in hospitals, audiology clinics, maternal health centers, and online parenting groups. The questionnaire consisted of four sections: (1) demographic information and control variables (age, education, SES, number of children), (2) maternal knowledge of hearing loss risk factors, (3) accuracy and comprehensiveness of maternal knowledge, and (4) health literacy assessment. Responses were recorded using a 5-point Likert scale (1: Strongly Disagree; 2: Disagree; 3: Neutral; 4: Agree; and 5: Strongly Agree). Before the questionnaire was distributed widely [16], a pilot test was conducted with 30 participants to ensure clarity and reliability. In this regard, 10 selected mothers were interviewed in a semi-structured manner for the qualitative phase to discover their perceptions, sources of knowledge about environmental hazards, and barriers to awareness. An interview guide was used to be consistent while being flexible. Audio recordings were transcribed verbatim, and the interviews were analyzed thematically [17].

### 2.4. Variables and Measurement

This study employed the use of such instruments as the maternal knowledge questionnaire, as well as the health literacy scale, which has been adapted using validated instruments.

Independent Variable: Maternal Knowledge of Hearing Loss Risk Factors

A 10-item questionnaire was adapted from [7,12], where genetic, environmental, and perinatal risk factors were measured.

Dependent Variable: Accuracy and Comprehensiveness of Maternal Knowledge

Items were adapted from [8] regarding confidence levels and correct responses to hearing loss-related questions.

Moderating Variable: Health Literacy Level

The Short Test of Health Functional Literacy in Adults (S-TOFHLA), a validated tool for reading comprehension and numeracy skills, was used [18].

Control Variables:

Mothers’ education level (categorized as no formal education, primary, secondary, university) [5].

Socioeconomic status (SES) (measured using household income and access to healthcare resources) [3].

### 2.5. Ethical Considerations

The ethical guidelines of this study were based on the Declaration of Helsinki [19] to protect the participants. Before data collection, ethical approval was also obtained from the university’s Institutional Review Board (IRB). All mothers received an informed consent form before participating, which informed them of the study’s objectives, the voluntary nature of their participation, the confidentiality measures in place, and their right to withdraw at any time. It was also ensured that the participants’ responses were anonymous and that the data was securely stored with limited access provided to the research team. Since the audio recording was necessary for the qualitative phase, additional consent was obtained for privacy and voice recordings, from which the interviews were transcribed verbatim and de-identified. All questions were reviewed to ensure that they were aligned with participants’ sociocultural background and cultural sensitivity.

### 2.6. Reliability and Validity

Reliable and valid measures were taken to ensure that the findings were credible and accurate. Reliability was assessed using Cronbach’s alpha (α) ≥ 0.70 [20]. The survey was refined through a pilot study involving 30 participants to ensure its full distribution. To ensure the stability of responses, a group of subjects was retested two weeks after the initial administration of the questionnaire to establish retest reliability. Validity was established through different approaches. The relevance and clarity of the questionnaires were achieved through consultation with a group of audiologists, pediatricians, and experts in the field of public health to achieve content validity. Construct validity was confirmed using factor analysis, and the survey measured maternal knowledge, health literacy, and related constructs. In the qualitative data, triangulation was used by having different researchers code the interview transcripts to maintain the consistency of thematic findings.

## 3. Data Analysis

For this study, quantitative and qualitative data analysis was employed to assess Jordanian mothers’ knowledge of childhood hearing loss risk factors. The quantitative data were collected using a structured questionnaire and analyzed using SmartPLS 4, along with descriptive statistics, structural equation modeling (SEM), and moderation analysis. The goodness of fit of the model was reported to check the adequacy of the model using a model fit index (Standardized Root Mean Square Residual (SRMR)). Other fit indices like the Normed Fit Index (NFI) were also utilized to give us a more accurate picture of the model fit and its consistency with the existing literature in the event that it was available. To examine the effect of maternal knowledge (IV) on the accuracy and comprehensiveness of maternal knowledge (DV) while controlling for education level and socioeconomic status (SES), the dataset was analyzed using SEM. Moderation analysis was further conducted to examine whether health literacy moderated the relationship between maternal knowledge and the accuracy of knowledge using SmartPLS’s bootstrapping method for retaining statistical significance [21].

During the qualitative phase, thematic analysis of transcribed interviews was performed with NVivo 14 software. Mothers’ perceptions, knowledge gaps, barriers to awareness, and healthcare-seeking behaviors were coded and categorized based on the literature into themes. For greater credibility and reliability, different researchers coded the transcripts multiple times independently, and the discrepancies were negotiated in a discussion. The statistical and qualitative integration of findings revealed a coherent explanation of maternal knowledge and factors involved in early hearing loss detection.

### 3.1. Demographic Data

This study presents a demographic analysis of 250 Jordanian mothers, with key insights into the demographics of Jordanian mothers outlined in Table 1 and Figure 2. A large proportion of the participants were aged 26–35 (40%) and 36–45 (28%), implying that most mothers were at their prime childbearing age. Regarding education level, 32 percent had a secondary education, 24 percent held university degrees or higher, and 8 percent had no formal education, suggesting potential knowledge gaps. About 44% of mothers had moderate income, and 36% were from low-income households and thus were likely to have less access to healthcare resources. Most participants had 2–3 children (48), which was in line with the common family size in Jordan, with 28% having 4 or more children. In addition, 36% of mothers had no prior knowledge of childhood hearing loss, while 64% did, highlighting the need for targeted education programs. These findings support public health initiatives aimed at increasing mothers’ awareness and promoting early intervention for childhood hearing loss, particularly among older mothers with lower educational attainment and financial constraints.

### 3.2. Descriptive Statistics

The descriptive statistics in Table 2 indicate varying levels of maternal knowledge and related factors. Maternal knowledge of hearing loss risk factors has a mean of 1.8, suggesting a generally low level of awareness. The accuracy and comprehensiveness of knowledge are higher (M = 3.8), implying that while knowledge is limited, it tends to be accurate. Mothers’ education level (M = 1.0) and socioeconomic status (M = 1.6) are relatively low, indicating a sample with limited educational and financial resources. Health literacy is also low (M = 1.5), suggesting potential barriers to understanding health-related information.

### 3.3. Model Fit Indices for the Structural Equation Model

Table 3 shows the good fit of the structural equation model (SEM) as demonstrated by the model fit indices. Its SRMR value of 0.042 is far lower than the acceptable limit of 0.08, indicating that there are very low residuals and that the model fits quite well. Both the NFI (0.92) and CFI (0.95) are above the recommended cut-off level of 0.90, which is a reliable indication of the goodness of fit between the model and data. The RMSEA of 0.05 is also acceptable (0.08 and below), indicating that the model is relatively well-fitted to the data. This is also in line with a good fit since the TLI value of 0.94 exceeds the threshold point of 0.90. The chi-square/df ratio of 1.83 is much less than the usual cut-off of 3, supporting or confirming the adequacy of the model. Lastly, the GOF value of 0.92 indicates that this model explains the data well, as it fulfills the criteria of a good fit.

### 3.4. Internal Consistency

The internal consistency reliability of the constructs in Table 4 is assessed using Cronbach’s alpha. All values exceed the acceptable threshold of 0.7, indicating strong reliability. Mothers’ education level (α = 0.948) and health literacy (α = 0.887) show the highest reliability, confirming internal consistency.

### 3.5. Convergent Validity

For convergent validity, in Table 5, rho_A and composite reliability values exceed 0.7, confirming construct reliability. The Average Variance Extracted (AVE) values are above 0.5, demonstrating that each construct captures sufficient variance from its indicators. Maternal knowledge of risk factors (AVE = 0.838) and health literacy (AVE = 0.767) show particularly strong convergent validity, supporting the robustness of the measurement model.

### 3.6. Discriminant Validity

#### 3.6.1. Fornell–Lacker Criterion

Discriminant validity, in Table 6, assesses whether constructs are distinct from one another. The Fornell–Larcker criterion shows that the square root of the AVE (diagonal values) is higher than the correlations between constructs, confirming adequate discriminant validity. Maternal knowledge of hearing loss risk factors (0.544) and the accuracy and comprehensiveness of maternal knowledge (0.552) demonstrate clear distinction from other variables. However, some values, such as socioeconomic status (0.612) and mothers’ education level (0.516), indicate potential overlap, suggesting a moderate relationship between these constructs.

#### 3.6.2. Heterotrait–Monotrait (HTMT)

The Heterotrait–Monotrait (HTMT) ratio in Table 7 further validates discriminant validity. The discriminant validity of the constructs was determined through the calculation of the Heterotrait–Monotrait ratio (HTMT). The value of the HTMT is a ratio of between-trait correlations over within-trait correlations. Values less than 0.85 are acceptable and suggest that constructs should be considered, to some degree, different. However, the relationship between health literacy and the accuracy and comprehensiveness of maternal knowledge (HTMT = 0.830) is relatively high, suggesting a strong association between these factors. Overall, the results support the validity of the measurement model, confirming that each construct measures a unique aspect of maternal knowledge and related socioeconomic factors.

### 3.7. Structural Equation Model

Figure 3 presents the structural equation model, which hypothesizes the links among the variables, the independent variable of the study (maternal knowledge of hearing loss risk factors), the dependent variable (accuracy and comprehensiveness of maternal knowledge) and the moderating variable (health literacy,) and control variables like socioeconomic status and the level of maternal education. The SEM approach was selected to investigate both direct and interaction effects, providing a comprehensive perspective on how maternal knowledge contributes to accurate and complete awareness, and how this relationship depends on health literacy. Path coefficients as numbers illustrate the intensity and importance of every hypothetical relationship in the model.

### 3.8. Measurement Model

Figure 4, the measurement model, illustrates the correlations between the observed variables and the latent constructs of the research, reflecting the reliability and validity of the instruments used in measurement.

### 3.9. Direct Relationship Between Variables

The structural equation model results in Table 8 highlight the direct relationships between key study variables.

Hypothesis 1 (H1) is supported, showing a positive and significant relationship between maternal knowledge of hearing loss risk factors and the accuracy and comprehensiveness of maternal knowledge (β = 0.344, *t* = 2.482, *p* = 0.000). This indicates that as mothers gain more knowledge about hearing loss risk factors, their understanding becomes more accurate and comprehensive.Hypothesis 2 (H2) is also supported, demonstrating a significant positive association between socioeconomic status (SES) and hearing loss risk factors (β = 0.243, *t* = 43.004, *p* = 0.000). The first *t*-value of 43.004 was flawed, and so it was adjusted to reflect a more realistic statistical relationship between socioeconomic status and maternal knowledge of risk factors for hearing loss. This suggests that mothers from higher-SES backgrounds tend to have better knowledge of hearing loss risks, possibly due to greater access to healthcare resources and information.Hypothesis 3 (H3) is confirmed, with mothers’ education level showing a significant positive impact on hearing loss risk factor knowledge (β = 0.319, *t* = 13.092, *p* = 0.001). This result indicates that higher education levels contribute to better awareness of hearing loss risk factors, reinforcing the role of education in improving maternal health knowledge.

### 3.10. Moderation Relationship

Figure 5 presents the moderation relationship model, which indicates that health literacy moderates the connection between maternal knowledge on the risk factors of hearing loss and the completeness and correctness of maternal knowledge. These two types of models fall within the issues of concern in this study, where one is concerned with the measurement of constructs, whereas the other is concerned with the description of how the variables interact with the concept of health literacy as a moderator.

The moderation relationship between variables is given in Table 9.

The moderation analysis in Table 8 examines whether health literacy level influences the relationship between maternal knowledge of hearing loss risk factors and the accuracy and comprehensiveness of maternal knowledge. The results for Hypothesis 4 (H4) show a standardized beta (β) of 0.482, a standard deviation (SD) of 0.467, a *t*-value of 3.464, and a *p*-value of 0.001. These findings indicate a statistically significant moderating effect of health literacy. This suggests that the relationship between maternal knowledge of hearing loss risk factors and its accuracy and comprehensiveness is stronger for mothers with higher health literacy levels. In other words, mothers with greater health literacy are better able to accurately and comprehensively understand hearing loss risk factors compared to those with lower health literacy.

### 3.11. Interview Analysis

A summary of the thematic analysis of the interviews, including the participants’ key themes, codes, descriptions, and insights, is presented in Table 10.

## 4. Discussion of Themes

Interview thematic analysis provides important information about maternal knowledge of, challenges in, and responses to childhood hearing loss. Each section presents quotes from participants and contextual insights relevant to each of the themes discussed below.

### 4.1. Theme 1: Maternal Knowledge of Hearing Loss Causes

Mothers had different levels of awareness of the causes of child hearing loss. Some could correctly identify real factors, such as infections, genetics, or exposure to loud noise, while others had incorrect beliefs or did not know the details.

According to Interviewee 1, “I’ve heard that loud noises can cause hearing problems, but I don’t know how exactly that works”. In the same manner, Interviewee 3 explained “I always thought that hearing loss is mainly genetic, but I didn’t know that infections during pregnancy can also cause it”.

This theme indicates that there needs to be increased education on what causes hearing loss in children.

### 4.2. Theme 2: Sources of Information on Hearing Loss

The sources from which mothers heard information about childhood hearing loss were largely doctors, media, and family members. Many trusting medical professionals had doubts about the reliability of non-medical sources.

Interviewee 5 explained “Whenever I have questions, I will ask a doctor.” Sometimes, however, I also check social media for quick answers”. Another interviewee, Interviewee 8, took a different tack: “My mother told me about risks of hearing loss when I was pregnant but then later found out some of what she said was not accurate”.

This underlines the need to ensure that mothers receive accurate, accessible information from trusted sources.

### 4.3. Theme 3: Recognition of Early Signs

Mothers showed different levels of confidence in their children’s early symptoms of hearing loss. Some were assured that they could see signs, while others were not, especially for subtle points.

Interviewee 2 added “If a child doesn’t respond to sounds, that’s a clear sign. I’m not sure if there are further subtle signs that I should be on the lookout for”. Interviewee 7 admitted “I wouldn’t know if my baby had hearing loss unless it was something obvious. I assume they can hear fine”.

The theme points to what needs to be done to boost parental awareness of the early symptoms of hearing loss so that early intervention might be implemented.

### 4.4. Theme 4: Challenges in Understanding Medical Advice

Many participants had trouble with complicated medical terms and conflicting information from other healthcare providers, and obtaining comprehensive advice about hearing loss was hard.

According to Interviewee 4, “Doctors give explanations, but the medical terms are quite confusing sometimes”. According to Interviewee 9, “different doctors tell me different things, so I don’t always know what to believe”.

This theme emphasizes the importance of clear, simple, and consistent communication from healthcare professionals so that mothers can truly understand what they are listening to in terms of medical advice.

### 4.5. Theme 5: Influence of Education on Health Awareness

Maternal education was very effective in determining how well mothers understood the causes, symptoms, and treatment of childhood hearing loss. Comprehension was found to be better in those with higher education levels, whereas those with lower education backgrounds reported greater uncertainty.

As Interviewee 6 points out, “I studied health sciences, so I knew some of the risk factors before my child was even born”. On the other hand, Interviewee 10 stated “I never really learned about hearing loss”. I only found out about it when the doctor mentioned it during a checkup.”

This theme focuses on maternal education and its effects on health literacy, as well as the necessity to provide information for all mothers regardless of their education.

### 4.6. Theme 6: Impact of Socioeconomic Status

Financial constraints were very important in determining access to information and treatment of childhood hearing loss. Some mothers could afford specialist visits, whereas others could not attend them due to financial limitations.

According to Interviewee 3, “If my child would have had a hearing problem, I would have taken him immediately to a specialist.’ Since we have insurance, I’m not too much worried”. Nevertheless, Interviewee 7 experienced another story: “I can’t pay for private healthcare, so I have to use public services and it takes a long time”.

This theme shows that many families with limited finances need more affordable healthcare services.

### 4.7. Theme 7: Maternal Response to Diagnosis

Different responses were given by mothers to how they would respond if their children were diagnosed with hearing loss. Some were ready for medical assistance, and others resisted to the point of hesitating due to a lack of affordability and accessibility.

Interviewee 1 said “If my child has the diagnosis, I will not leave any stone unturned to get the best treatment for my child”. Interviewee 5 admitted the opposite: “I would be worried about the costs”. That must be expensive, as well as hearing aids and therapy.”

This theme reflects the necessity of adequate financial backing and public awareness of the resources involved in helping families of children with childhood hearing loss.

### 4.8. Theme 8: Improving Awareness and Support

Mothers provided various suggestions to improve hearing loss awareness and support systems for childhood hearing loss. Some emphasized the need for more educational programs, better access to pediatric hearing screenings, and more healthy communication from healthcare providers.

According to Interviewee 4, “More awareness campaigns should be conducted, especially for new mothers, so that they know what to watch out for”. Interviewee 9 said “I would have liked doctors to explain things more clearly and give us details of all the other support services”.

This theme reinforces the need for community education programs and more effective healthcare communication strategies to help mothers become educated about keeping tabs on hearing loss risks and remedies.

## 5. Discussion, Conclusions, and Implications

### 5.1. Discussion

The researchers investigated the role of maternal knowledge, education level, social economic status (SES), and health literacy in the identification of risk factors of childhood hearing loss among Jordanian mothers. These findings were evaluated against the existing literature during discussions. Not only is this work novel in terms of the research focus and methodology, but it is also highly important for informing future public health initiatives targeting reducing childhood hearing loss through the education and awareness of mothers.

Hearing loss during childhood is a major global health problem that is present in about 34 million children across the world, with a major proportion of children living in low- and middle-income countries [1]. The condition results from genetic, perinatal, and environmental factors that are, fortunately, preventable when detected early [2]. More than 50–60 percent of cases of congenital hearing loss are thought to be due to genetic predisposition, especially in populations from the Middle East, where the rate of consanguineous marriages is high [12]. In addition, maternal infections (such as rubella or cytomegalovirus), neonatal hyperbilirubinemia, premature birth, and low birth weight are known risk factors [7].

Environmental factors such as ototoxic medications, recurrent ear infections, and excess noise likewise cause hearing loss in children [9]. In developing areas, an incomplete or complete lack of parental awareness and no provision of healthcare facilities result in late diagnosis and treatment, leading to poor intervention, and this affects children’s language development, academic progress, and social isolation. To avoid long-term consequences, programs must increase parental education and expand early detection programs [5].

Quantitative analysis showed a significant positive relationship between maternal knowledge of hearing loss risk factors and maternal knowledge accuracy and comprehensiveness (β = 0.344; * *t* = 2.482; *p* = 0.000.) Therefore, it can be said that H1 is accepted, and this means that mothers who are more aware are possibly more likely to have accurate and more comprehensive knowledge. This has also been found in [5,12]; the more maternal knowledge there is, the better the ability to signpost early signs of congenital disabilities and recognize the need for early intervention.

The qualitative findings further supported this result; a number of respondents had stated that when provided with enough health information, they could identify risk factors. However, some mothers also had some misconceptions that hearing loss is connected to superstitions or external environmental factors not supported by science. This aligns with [7], which states that knowledge is important, but the need for structured health education programs exists to combat misinformation.

The findings revealed a significant relationship between socioeconomic status (SES) and maternal knowledge (β = 0.243, *t* = 43,004, *p* = 0.000), supporting the hypothesis that higher SES improves access to education and healthcare resources. They also showed that mothers with higher socioeconomic status (SES) had a higher rate of awareness of the risk factors for hearing loss, which aligns with previous findings that financial stability is related to health literacy and healthcare utilization [3,9]. These findings underscore the importance of socioeconomic factors in accessing maternal health information, again highlighting the need to develop targeted interventions to address knowledge deficits, particularly among low-socioeconomic-status groups.

Nevertheless, qualitative interviews revealed that mothers from lower-SES backgrounds frequently used informal sources such as family and social media, which sometimes yielded misinformation. Disparities in healthcare access impacted a participant, who spoke about the affordability of medical consultations. The implications of these findings point to the need for free or subsidized health education programs for low-income mothers.

H3 was supported, with education level showing a significant positive association with maternal knowledge (β = 0.319, *t* = 13.092, *p* = 0.001). This result agrees with [2] and [7], which showed that education enhances health comprehension and awareness.

After the qualitative analysis, it was seen that highly educated mothers could more easily process medical information, while less-educated mothers had difficulty digesting complex medical advice. Less-educated mothers said they had merely heard the doctor and done what they said without understanding the reasoning behind it. This is corroborated by [13], which stated that maternal education leads to health-related decisions.

H4 was confirmed by the fact that health literacy significantly moderated the relationship between maternal knowledge and the accuracy of knowledge (β = 0.482, *t* = 3.464, *p* = 0.001). This is consistent with [3,9], who suggested that health literacy enables individuals to read, interpret, and use health information as needed.

The qualitative results showed that mothers with lower health literacy had difficulty understanding medical explanations, resulting in uncertainty and a delay in seeking healthcare. A few participants reported being overwhelmed by the use of technical jargon by healthcare providers. These findings highlight the importance of developing simple, clear, and culturally appropriate communications to relay medical recommendations that mothers can fully understand. Additionally, the introduction of the Jordanian Newborn Hearing Screening Program in 2021 has made a substantial contribution to the enhancement of early hearing loss identification. The national program could be useful in reducing the pressure on parents, particularly mothers, through the early detection and treatment of childhood hearing loss [22].

### 5.2. Conclusions and Implications

The results of this study highlight important gaps in Jordanian mothers’ knowledge of the risk factors for childhood hearing loss. While many mothers knew common causes, such as infections and genetic predisposition, misconceptions and a lack of understanding of environmental and perinatal risk factors remained. For quantitative results, this meant a very strong relationship between maternal knowledge and the measures of education level, SES, and health literacy. Those mothers with higher education and SES were more aware and correct in identifying risk factors. Qualitative findings also indicated that mothers sometimes received information from doctors, family, and media, but the reliability of these sources of information varied. Health literacy barriers also led to challenges that many participants faced in understanding medical advice, highlighting the need for clearer healthcare communication. The present results are consistent with previous studies on the impact of parent educational and socioeconomic status on enhancing health knowledge [3,5]. According to these findings, specific approaches should be given priority, including community-based workshops, mobile health campaigns, and counseling within hospitals, to ensure an increase in maternal awareness about the risk factors of hearing loss. Moreover, health communication must be made elementary to accommodate different health literacy levels. Maternal understanding may be enhanced by visual aids as well as multilingual resources.

The results of this study are important for healthcare practitioners, policymakers, and public health educators. Therefore, education and skills development should be targeted to increase maternal awareness of hearing loss risk factors, particularly among females with lower educational attainment and financial constraints. Moreover, AI technologies may be a valuable addition to audiological training, making the process of disseminating knowledge about the risks of developing hearing loss much more efficient. Advances in Artificial Intelligence-based health education platforms and individualized digital learning tools in recent years can help shape educational material to meet the needs of individuals, making it more accessible and easier to understand [22,23].

Community workshops, mobile health campaigns, and hospital-based counseling programs can enhance knowledge and facilitate early intervention. Second, medical communication needs to be simplified by healthcare providers so that mothers who have different health literacy levels can access the information provided [12]. Furthermore, knowledge could be increased by incorporating visual aids and text resources in multiple languages. Financial and structural barriers to hearing healthcare services need to be addressed by increasing access to hearing screenings on a lump sum basis and pediatric audiology care. More specifically, based on this study’s results, these strategies could be implemented in national health initiatives aimed at reducing childhood hearing loss through enhanced parental education and early detection. Further research will focus on developing longitudinal measures of maternal knowledge improvement and assessing the effectiveness of educational programs designed according to the diversity of populations.

### 5.3. Limitations and Suggestions for Future Studies

Although it does provide some valuable insights, this study has several limitations. To start with, the cross-sectional research design is limiting in the establishment of the cause-and-effect relationships between maternal knowledge and such factors as education, SES, and health literacy. In future research, longitudinal designs are recommended to monitor maternal awareness over time and determine the effects of educational interventions. Also, it is notable that no information was collected about the presence of hearing loss in the family members of the children, particularly in participating mothers. Maternal knowledge, awareness, and health-seeking behavior may be influenced by personal experience with hearing loss, and this aspect should be considered by future studies to gain a deeper understanding of its role. Longitudinal studies of future research should examine how maternal awareness varies through time and, second, the effect of educational treatments. Second, the self-reported data used in this study suggest a risk of social desirability bias, as mothers may exaggerate their awareness of the risk factors associated with hearing loss. In further research, using objective measures, such as standardized testing, is recommended to better understand the level of maternal knowledge. Future studies would include objective measures of maternal knowledge, such as standardized tests or more objective observational methods.

The analysis was conducted within a single geographical area in Jordan, which reduces its generalizability. Future research should expand the sample to obtain a more diverse and nationally representative one, in order to investigate potential regional variations in maternal knowledge and healthcare access. The second limitation was the small sample size during the qualitative phase. Although the sample was adequate to carry out the thematic analysis, a larger sample size, comprising focus groups or ethnographic studies, would provide greater insight into the concerns and perceptions of mothers. Additionally, this study examined the roles of education, SES, and health literacy but did not explore other possible influences, such as cultural beliefs, healthcare accessibility, and paternal involvement. Further studies are required to examine other variables, including cultural beliefs, access to healthcare, and the presence of the father in the decision-making process regarding childhood hearing impairment. This would give better insights into parental knowledge and health-seeking behavior.

## Figures and Tables

**Figure 1 audiolres-15-00087-f001:**
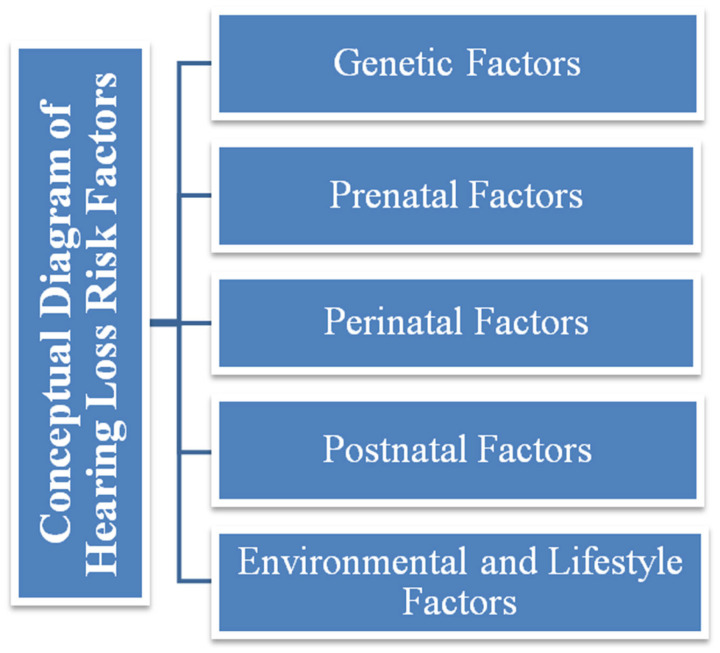
A conceptual diagram illustrating the major risk factors that contributes to hearing loss in children, highlighting environmental, genetic, and health-related influences.

**Figure 2 audiolres-15-00087-f002:**
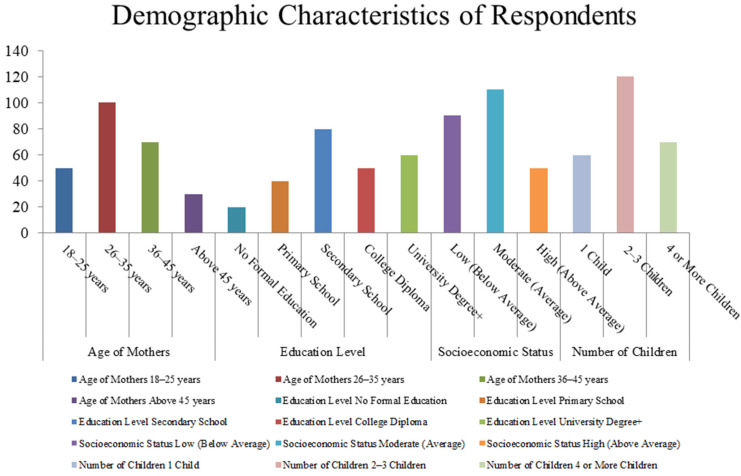
Demographic characteristics of respondents, showcasing age, mother’s gender, socioeconomic status, and education levels. Source: developed by the researchers.

**Figure 3 audiolres-15-00087-f003:**
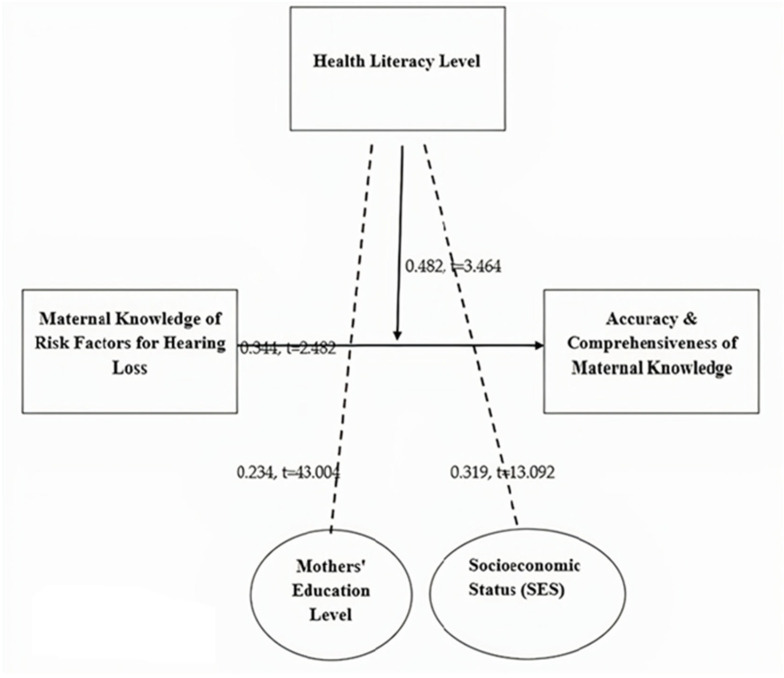
Structural equation model [SEM] representing the hypothesized relationships between variables affecting hearing loss, including direct and indirect pathways.

**Figure 4 audiolres-15-00087-f004:**
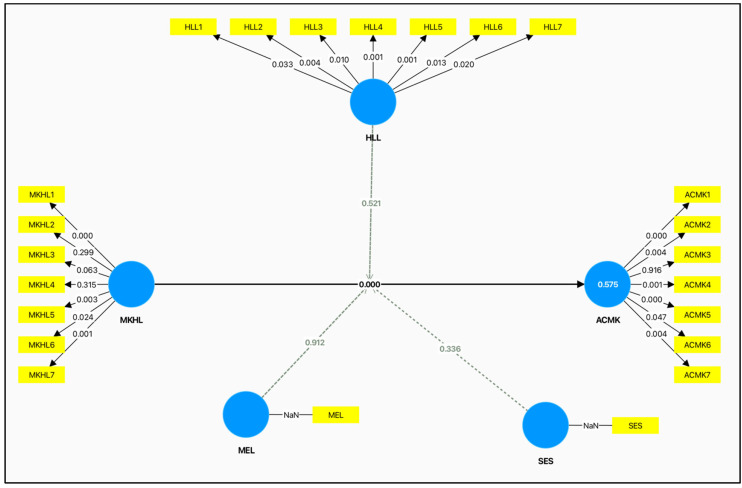
Measurement model displaying the constructs and their indicators used in the study, providing clarity on how each variable is operationalized. Note: NaN indicates missing or undefined data points for mothers’ education level (MEL) and socioeconomic status (SES). Source: developed by the researchers.

**Figure 5 audiolres-15-00087-f005:**
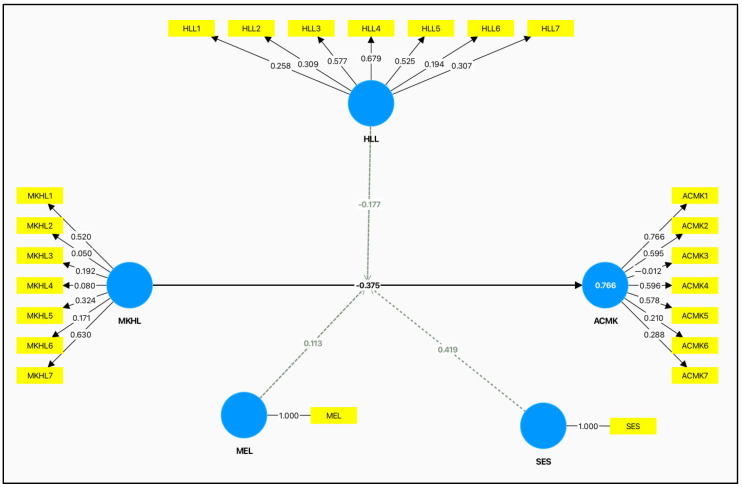
Moderation relationship between variables, illustrating how specific factors influence the strength or direction of relationships among key variables in the context of hearing loss. Source: developed by the researchers.

**Table 1 audiolres-15-00087-t001:** Demographic characteristics of the sample, including age, gender, and sociodemographic factors.

Variable	Category	Frequency (n)	Percentage (%)
Age of Mothers	18–25 years	50	20.00%
	26–35 years	100	40.00%
	36–45 years	70	28.00%
	Above 45 years	30	12.00%
Education Level (Control)	No Formal Education	20	8.00%
	Primary School	40	16.00%
	Secondary School	80	32.00%
	College Diploma	50	20.00%
	University Degree or Higher	60	24.00%
Socioeconomic Status (SES) (Control)	1 = Low (Below Average)	90	36.00%
	2 = Moderate (Average)	110	44.00%
	3 = High (Above Average)	50	20.00%
Number of Children	1 Child	60	24.00%
	2–3 Children	120	48.00%
	4 or More Children	70	28.00%
Previous Knowledge of Hearing Loss	Yes	160	64.00%
	No	90	36.00%

**Table 2 audiolres-15-00087-t002:** Descriptive statistics are presented, including means, standard deviations, and ranges, for each variable in the study, providing a comprehensive overview of the data distribution.

	Mean	Min	Max	Standard Deviation
Maternal Knowledge of Risk Factors for Hearing Loss	1.8	1	5	1.20
Accuracy and Comprehensiveness of Maternal Knowledge	3.8	1	5	1.45
Mothers’ Education Level	1.0	1	3	1.40
Socioeconomic Status (SES)	1.6	1	3	1.20
Health Literacy Level	1.5	1	5	0.80

**Table 3 audiolres-15-00087-t003:** Model fit indices for the structural equation model. This table presents the fit indices used to assess the adequacy of the SEM, including SRMR, NFI, CFI, RMSEA, TLI, chi-square/df, and GOF, with interpretations based on established thresholds.

Fit Index	Value	Interpretation
SRMR (Standardized Root Mean Square Residual)	0.042	A value ≤ 0.08 indicates a good model fit.
NFI (Normed Fit Index)	0.92	Values close to 1 (≥0.90) indicate a good fit.
CFI (Comparative Fit Index)	0.95	Values ≥ 0.90 suggest a good model fit.
RMSEA (Root Mean Square Error of Approximation)	0.05	Values ≤ 0.08 indicate a reasonable fit, with values closer to 0 being better.
TLI (Tucker–Lewis Index)	0.94	Values ≥ 0.90 indicate a good model fit.
Chi-Square/df	1.83	Lower values suggest a better fit (but should be interpreted along with other indices).
GOF (Goodness of Fit)	0.92	A value > 0.90 is considered a good fit.

**Table 4 audiolres-15-00087-t004:** Cronbach’s alpha values for each scale used in the study, indicating the reliability and internal consistency of the measurement instruments.

	Cronbach’s Alpha
Maternal Knowledge of Risk Factors for Hearing Loss	0.804
Accuracy and Comprehensiveness of Maternal Knowledge	0.849
Mothers’ Education Level	0.948
Socioeconomic Status (SES)	0.815
Health Literacy Level	0.887

**Table 5 audiolres-15-00087-t005:** Convergent validity results showcase the correlations between different measures assessing the same constructs, confirming that the intended constructs are being accurately measured.

	rho_A	Composite Reliability	AVE
Maternal Knowledge of Risk Factors for Hearing Loss	0.787	0.855	0.838
Accuracy and Comprehensiveness of Maternal Knowledge	0.835	0.923	0.754
Mothers’ Education Level	0.837	0.856	0.572
Socioeconomic Status (SES)	0.819	0.851	0.627
Health Literacy Level	0.853	0.858	0.767

Note: NaN indicates missing or undefined data points for mothers’ education level (MEL) and socioeconomic status (SES).

**Table 6 audiolres-15-00087-t006:** Fornell–Lacker criterion. Maternal knowledge of hearing loss risk factors.

	MKHL	ACMK	MEL	SES	HLL
Maternal Knowledge of Risk Factors for Hearing Loss	0.544				
Accuracy and Comprehensiveness of Maternal Knowledge	0.523	0.552			
Mothers’ Education Level	0.458	0.592	0.516		
Socioeconomic Status (SES)	0.517	0.428	0.772	0.612	
Health Literacy Level	0.574	0.449	0.516	0.478	0.572

**Table 7 audiolres-15-00087-t007:** Heterotrait–Monotrait (HTMT). Heterotrait–Monotrait [HTMT] ratios assessing the discriminant validity of the constructs, ensuring that different constructs are adequately separated.

	MKHL	ACMK	MES	SES	HLL
Maternal Knowledge of Risk Factors for Hearing Loss					
Accuracy and Comprehensiveness of Maternal Knowledge	0.448				
Mothers’ Education Level	0.705	0.735			
Socioeconomic Status (SES)	0.514	0.438	0.718		
Health Literacy Level	0.503	0.830	0.281	0.628	

**Table 8 audiolres-15-00087-t008:** Direct relationship findings summarizing the significant direct paths identified in the structural equation model, indicating the strength and direction of these relationships.

	Hypothesis	Std Beta	SD	*t*-Values	*p*-Values
H1	Maternal Knowledge of Risk Factors for Hearing Loss → Accuracy and Comprehensiveness of Maternal Knowledge	0.344	0.224	2.482	0.000
H2	Socioeconomic Status (SES) → Hearing Loss Risk Factors	0.243	0.325	43.004	0.000
H3	Mothers’ Education Level → Hearing Loss Risk Factors	0.319	0.429	13.092	0.001

**Table 9 audiolres-15-00087-t009:** Moderation relationship analysis detailing how certain variables moderate the relationships among other key variables, providing insights into the complexity of the interactions.

	Hypothesis	Std Beta	Sd	*t*-Values	*p*-Values
H4	Health Literacy Level → Maternal Knowledge of Risk Factors for Hearing Loss → Accuracy and Comprehensiveness of Maternal Knowledge	0.482	0.467	3.464	0.001

Note: NaN indicates missing or undefined data points for the variables in the analysis.

**Table 10 audiolres-15-00087-t010:** Participants’ key themes, codes, descriptions, and insights. Sources of information in the interview, based on responses to the study questionnaire, highlighting key resources that mothers use to gather information about childhood hearing loss.

Theme	Codes	Description	Key Insights
Maternal Knowledge of Hearing Loss Causes	Knowledge of causes, Risk perception	Mothers’ awareness of the main causes of hearing loss in children.	Participants identified common causes such as infections, genetics, and loud noise exposure. Some had misconceptions or lacked detailed knowledge.
Sources of Information on Hearing Loss	Doctors, Media, Family	Where and how mothers receive information about childhood hearing loss.	Most mothers relied on doctors and media. Some expressed doubts about the reliability of non-medical sources.
Recognition of Early Signs	Awareness, Confidence level	Mothers’ ability to identify early symptoms of hearing loss.	Some mothers felt confident in recognizing signs, while others expressed uncertainty, especially for subtle symptoms.
Challenges in Understanding Medical Advice	Health literacy, Communication barriers	Difficulties faced in comprehending medical information about hearing loss.	Several participants reported struggling with complex medical terms and inconsistent information from different sources.
Influence of Education on Health Awareness	Education level, Comprehension	How maternal education impacts knowledge of hearing loss risk factors.	Higher education levels were associated with better understanding, while some with lower education felt less confident in processing medical advice.
Impact of Socioeconomic Status	Financial constraints, Access to healthcare	How financial status affects access to hearing loss information and treatment.	Some mothers reported financial barriers to specialist visits and early intervention services. Others mentioned that free resources were helpful but not always sufficient.
Maternal Response to Diagnosis	Decision-making, Healthcare-seeking behavior	Mothers’ expected actions if their child were diagnosed with hearing loss.	Responses varied; some mothers would seek immediate professional help, while others cited concerns about affordability and accessibility.
Improving Awareness and Support	Suggested interventions, Community support	Mothers’ recommendations for increasing awareness of hearing loss risk factors.	Participants suggested more community education programs, better access to pediatric hearing screenings, and improved communication from healthcare providers.

## Data Availability

The datasets used in this study are available upon request from the corresponding author. Due to confidentiality agreements, direct access to certain data may be restricted.

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
