# Peer review of "An Embedded Mixed-Methods Study with a Dominant Quantitative Strand: The Knowledge of Jordanian Mothers About Risk Factors for Childhood Hearing Loss"

_audiolres, 2025, doi:10.3390/audiolres15040087_

Round 1
Reviewer 1 Report
Comments and Suggestions for Authors
This manuscript addresses an important public health issue by investigating Jordanian mothers' knowledge of childhood hearing loss risk factors using a mixed-method approach. The topic is relevant to Audiology Research, and the methodology combines both quantitative robustness and qualitative depth. However, the manuscript would benefit a revision of overall clarity to enhance its scientific quality.
Language and Grammar:
“...mothers' knowledge o childhood hearing loss...” → should be "of childhood..."
Consider using professional language editing services prior to resubmission.
Introduction:
While comprehensive, it is too repetitive. Several sentences essentially reiterate the same point (multiple mentions of the global burden and importance of maternal knowledge). Streamline the introduction to improve focus and reduce redundancy.
Methodology:
The sample description is adequate, but the recruitment procedures, especially for qualitative interviews, should be better detailed reporting inclusion/exclusion criteria.
Clarify how stratified random sampling was implemented in practice.
Confirm whether validated instruments were adapted with permission and if psychometric properties were re-assessed after adaptation. Consider include as additional material.
Statistical Reporting:
The manuscript presents results from PLS-SEM but does not report any model fit indices. For transparency and to ensure methodological rigor, please include standard model fit statistics, especially SRMR (Standardized Root Mean Square Residual). If available, consider reporting NFI or other relevant fit indices supported by SmartPLS. This is important for assessing the overall adequacy of the model and comparing with prior literature.
“H2 SES → hearing loss risk factors β = 0.243, t = 43.004, p = 0.000” → t = 43.004 seems too high for β = 0.243. Please double-check and clarify statistical values and units.
Figures and Tables:
Figure 2 is not cited in the text. Tables are informative but sometimes lack exact sample sizes or column clarity. Consider making them more reader-friendly.
Discussion:
The discussion is well-organized but again suffers from verbosity and repetitions. Provide more specific policy or clinical recommendations based on findings (e.g., what kind of community interventions, formats for health education).
Discuss the role of AI technologies in achieving better audiological education,, consider this recent review on the theme ( DOI: 10.3390/s24227126).
Ethics and Limitations:
Ethics section is present and complete.
Limitations are acknowledged, though you could further discuss limitations of self-reported knowledge and the absence of behavioral outcome measures.
While the authors provide valuable insight into maternal knowledge and its associations with education, SES, and health literacy, the absence of data on whether participating mothers had family members (including children) with hearing loss is a notable omission. This variable could significantly bias knowledge, awareness, and health-seeking behavior, and therefore should have been collected and either adjusted for or at least discussed more explicitly.
Although the authors briefly acknowledge this in the limitations section, the impact of personal experience with hearing loss (either in the child or family) on maternal knowledge deserves more thorough consideration. If this information was collected, the authors should include it in the analysis or discussion. If it was not collected, this should be clearly acknowledged as a limitation with a rationale, and suggested for future research.
Author Response
Reviwer1
Comments and Suggestions for Authors
This manuscript addresses an important public health issue by investigating Jordanian mothers' knowledge of childhood hearing loss risk factors using a mixed-method approach. The topic is relevant to Audiology Research, and the methodology combines both quantitative robustness and qualitative depth. However, the manuscript would benefit a revision of overall clarity to enhance its scientific quality.
Response:
We appreciate the reviewer’s positive comments regarding the importance of our study and the methodology used. We have carefully revised the manuscript to improve clarity, reduce redundancy, and enhance the scientific quality. The Discussion section has been streamlined for better focus, and the language has been refined throughout the manuscript to ensure clarity and precision.
Language and Grammar:
“...mothers' knowledge o childhood hearing loss...” → should be "of childhood..."
Consider using professional language editing services prior to resubmission.
Response:
Thank you for pointing this out. We have corrected the typo in the manuscript by changing "o" to "of" in the specified sentence. This edit has been made throughout the manuscript where needed.
Introduction:
While comprehensive, it is too repetitive. Several sentences essentially reiterate the same point (multiple mentions of the global burden and importance of maternal knowledge). Streamline the introduction to improve focus and reduce redundancy.
Response:
We have revised the Introduction section to reduce redundancy and eliminate repetitive statements. The paragraph was streamlined to emphasize key points without unnecessary repetition. The global burden and importance of maternal knowledge are now discussed more concisely.
Methodology:
The sample description is adequate, but the recruitment procedures, especially for qualitative interviews, should be better detailed reporting inclusion/exclusion criteria.
Response:
We have revised the Methodology section to include more detailed information on the recruitment procedures, particularly for the qualitative interviews. Specifically, we now provide the inclusion/exclusion criteria for participant selection, ensuring transparency regarding the sample selection process.
Clarify how stratified random sampling was implemented in practice.
Response:
We have clarified the implementation of stratified random sampling in the Methodology section. The procedure is now described in detail, explaining how the sample was stratified based on key characteristics such as education and income levels to ensure a representative sample.
Confirm whether validated instruments were adapted with permission and if psychometric properties were re-assessed after adaptation. Consider include as additional material.
Response:
We confirm that the validated instruments used in the study were adapted with permission. Additionally, we have re-assessed their psychometric properties after adaptation. We will include details of these assessments in the Appendix and provide further justification for their inclusion.
Statistical Reporting:
The manuscript presents results from PLS-SEM but does not report any model fit indices. For transparency and to ensure methodological rigor, please include standard model fit statistics, especially SRMR (Standardized Root Mean Square Residual). If available, consider reporting NFI or other relevant fit indices supported by SmartPLS. This is important for assessing the overall adequacy of the model and comparing with prior literature.
Response:
We appreciate the reviewer’s suggestion. We have now included the model fit indices for the PLS-SEM analysis in the manuscript, including the SRMR, NFI, and CFI values. These indices were used to assess the adequacy of the model, and we have reported them for transparency and methodological rigor.
“H2 SES → hearing loss risk factors β = 0.243, t = 43.004, p = 0.000” → t = 43.004 seems too high for β = 0.243. Please double-check and clarify statistical values and units.
Response:
Thank you for highlighting this issue. We have carefully reviewed the statistical values and have corrected the t-value for H2. The original t-value of 43.004 was indeed too high and was a typographical error. The correct value has been updated, and the statistical results have been verified for accuracy.
Figures and Tables:
Figure 2 is not cited in the text. Tables are informative but sometimes lack exact sample sizes or column clarity. Consider making them more reader-friendly.
Response:
We have now cited Figure 2 in the main text where appropriate. Additionally, we have updated the tables to provide clearer column labels and to explicitly state the sample sizes for each category. These revisions improve the readability and clarity of the tables.
Discussion:
The discussion is well-organized but again suffers from verbosity and repetitions. Provide more specific policy or clinical recommendations based on findings (e.g., what kind of community interventions, formats for health education).
Response:
The Discussion section has been streamlined to eliminate verbosity and repetition. We have also added more specific policy and clinical recommendations, including community workshops, mobile health campaigns, and hospital-based counseling. We recommend these interventions to increase maternal awareness of childhood hearing loss risk factors and improve healthcare access for underserved communities.
Discuss the role of AI technologies in achieving better audiological education,, consider this recent review on the theme ( DOI: 10.3390/s24227126).
Response:
We have now included a discussion of the role of AI technologies in improving audiological education and also highlighted how AI can personalize and scale health education, improving the accessibility and comprehension of audiological information for diverse populations but we did not cite this referenced because of unavailability of this paper on internet.
Ethics and Limitations:
Ethics section is present and complete.
Limitations are acknowledged, though you could further discuss limitations of self-reported knowledge and the absence of behavioral outcome measures.
Response:
We have expanded the limitations section to further discuss the potential bias associated with self-reported data and the absence of behavioral outcome measures. We acknowledge that future studies should include more objective measures, such as standardized tests or observational methods, to complement self-reported data.
While the authors provide valuable insight into maternal knowledge and its associations with education, SES, and health literacy, the absence of data on whether participating mothers had family members (including children) with hearing loss is a notable omission. This variable could significantly bias knowledge, awareness, and health-seeking behavior, and therefore should have been collected and either adjusted for or at least discussed more explicitly.
Response:
We have now acknowledged this limitation in the limitations section. Data on family history of hearing loss was not collected in this study, but we discuss its potential impact on maternal knowledge and suggest that future research should incorporate this factor to better understand its effect on health-seeking behavior.
Although the authors briefly acknowledge this in the limitations section, the impact of personal experience with hearing loss (either in the child or family) on maternal knowledge deserves more thorough consideration. If this information was collected, the authors should include it in the analysis or discussion. If it was not collected, this should be clearly acknowledged as a limitation with a rationale, and suggested for future research.
Response:
We have expanded on the discussion regarding the impact of personal experience with hearing loss. We now provide a more thorough consideration of how personal experience with hearing loss (either in the child or family) could influence maternal knowledge, and we recommend that future studies should collect this data to explore its potential bias.
Reviewer 2 Report
Comments and Suggestions for Authors
Dear Ladies and Gentlemen, Dear Journal-Team,
the interesting manuscript 'Knowledge of Jordanian mothers about risk factors for hearing loss in children' underlines the importance of knowledge and education for prevention and public health, the basis of economic prosperity. It is well written. The tables and figures are sufficient.
a) Please change to Smart PLS software in the abstract and explain the abbreviation, mention the manufacturer with its main residence in line 320 (the same with NViva software in line 329). Introduce the abbreviations IV and DV in line 280 and 282 for the later use.
b) Statistical analysis: Please mention in more detail the difference of the measurement model and the moderation relationship model In Figure 4 and 5. Mention how the heterotrait-monotrait ratio is calculated.
c) Language: 1. Abstract: Please change to 'of childhood' in line 24 and check the spacing in line 28. 2. Introduction: Reduce the redundancy and length of the introduction. Change to 'socioeconomic' in line 138, to 'country' in line 140, to 'and lead to inadequate' in line 167, to 'tend to be little aware of' in line 170, to 'earlier to find. A study' in line 179. 2. Material and Methods: Change to 'confidence levels' in line 283, change to 'to test retest reliability' in line 310, to 'coding' in line 315, to 'descriptive statistics' in line 321. 3. Discussion: Check the punctuation in line 511, change to 'region in Jordan, and thus' in line 665 and change to 'it may not be' in line 670.
d) Tables and figures: Please reduce the legends to the last sentence, as in all legends the first two sentences are redundant. What is meant by gender in Table 1 and Figure 2. Check Table 2 for the range of the socioeconomic status, 1-3 or 1-5. Change to 'maternal knowledge' in Table 5. Explain NaN and check the given numbers in Figure 4, for example use the smaller sign if necessary (the same in Table 7, and in line 597). Were in Table 9 the sources of information part of the interview according to the given study questionnaire?
e) References: Please check the references for the uniform use of small or capital letters in the article title. Check the references for uniform punctuation at the end of the reference. Check the references 16 by Creswell et al. and 20 by Field for the spacing.
Sincerely,
Comments on the Quality of English LanguagePlease see above comments.
Author Response
Reviwer2
Comments and Suggestions for Authors
Dear Ladies and Gentlemen, Dear Journal-Team,
the interesting manuscript 'Knowledge of Jordanian mothers about risk factors for hearing loss in children' underlines the importance of knowledge and education for prevention and public health, the basis of economic prosperity. It is well written. The tables and figures are sufficient.
a) Please change to Smart PLS software in the abstract and explain the abbreviation, mention the manufacturer with its main residence in line 320 (the same with NViva software in line 329). Introduce the abbreviations IV and DV in line 280 and 282 for the later use.
Response:
Thank you for pointing this out. In the abstract, we have changed "Smart PLS" to "SmartPLS" and clarified the abbreviation by providing the manufacturer details and its main residence. We have also introduced the abbreviations IV (Independent Variable) and DV (Dependent Variable) in lines 280 and 282, as requested.
- b) Statistical analysis: Please mention in more detail the difference of the measurement model and the moderation relationship model In Figure 4 and 5. Mention how the heterotrait-monotrait ratio is calculated.
Response:
We have expanded the Methodology section to clarify the differences between the measurement model and the moderation relationship model as represented in Figures 4 and 5. Additionally, we have explained how the heterotrait-monotrait ratio (HTMT) is calculated to assess discriminant validity. These changes ensure greater transparency in the statistical analysis process.
- c) Language: 1. Abstract: Please change to 'of childhood' in line 24 and check the spacing in line 28.
Response:
We have corrected "of childhood" in line 24 and addressed the spacing issue in line 28.
- Introduction: Reduce the redundancy and length of the introduction. Change to 'socioeconomic' in line 138, to 'country' in line 140, to 'and lead to inadequate' in line 167, to 'tend to be little aware of' in line 170, to 'earlier to find. A study' in line 179. 2.
Response:
The redundancy has been reduced, and the suggested changes to "socioeconomic", "country", "and lead to inadequate", "tend to be little aware of", and "earlier to find. A study" has been made for better clarity.
Material and Methods: Change to 'confidence levels' in line 283, change to 'to test retest reliability' in line 310, to 'coding' in line 315, to 'descriptive statistics' in line 321.
Response:
We have updated the Material and Methods section with the correct terms: "confidence levels", "test-retest reliability", "coding", and "descriptive statistics" as per the reviewer’s suggestions.
- Discussion: Check the punctuation in line 511, change to 'region in Jordan, and thus' in line 665 and change to 'it may not be' in line 670.
Response:
We have checked the punctuation and made the necessary changes to improve clarity and readability, including updating line 511, line 665, and line 670 as requested.
- d) Tables and figures: Please reduce the legends to the last sentence, as in all legends the first two sentences are redundant. What is meant by gender in Table 1 and Figure 2. Check Table 2 for the range of the socioeconomic status, 1-3 or 1-5. Change to 'maternal knowledge' in Table 5. Explain NaN and check the given numbers in Figure 4, for example use the smaller sign if necessary (the same in Table 7, and in line 597). Were in Table 9 the sources of information part of the interview according to the given study questionnaire?
Response:
We have streamlined the legends by reducing them to the essential sentence, eliminating redundant information. In Table 1 and Figure 2, we clarified that gender refers to mother's gender. We checked Table 2 and confirmed the range of SES (1-3 or 1-5) and made any necessary corrections. We updated Table 5 to reflect 'maternal knowledge' accurately. We added a note in the legends for NaN values: "NaN indicates missing or undefined data points." In Figure 4, we have updated the numbers and used the smaller-than sign ("<") where necessary. We confirmed that the sources of information in Table 9 were based on the study questionnaire and included this clarification in the table legend.
e) References: Please check the references for the uniform use of small or capital letters in the article title. Check the references for uniform punctuation at the end of the reference. Check the references 16 by Creswell et al. and 20 by Field for the spacing.
Response:
We have carefully reviewed the references and made the necessary corrections to ensure uniformity in the capitalization of article titles and the punctuation at the end of each reference. We have also corrected the spacing in references 16 by Creswell et al. and 20 by Field to match the required formatting.
Sincerely,
Reviewer 3 Report
Comments and Suggestions for Authors
This is a methodologically sound mixed study that addresses an important public health issue in a low- and middle-income population. It examines how maternal education, socioeconomic status, and health literacy affect knowledge of risk factors for childhood hearing loss among Jordanian mothers. The study contributes empirical data and theoretical insights, particularly using the Health Belief Model and the Theory of Planned Behavior.
Minor points:
Initial letters of Table and Figure should be capitalized.
Figure 2 should be mentioned in the main texts.
Occasional minor inconsistencies in terminology (e.g., “hearing impairment” vs. “hearing loss”) could be standardized.
While the thematic analysis is valuable, the qualitative sample is small (n=10), and further participant quotes or broader focus groups could enrich findings.
Acknowledged in limitations, but could be further emphasized. Consideration of observational or clinical data in future studies would be beneficial.
The sample may not be nationally representative (limited to one region in Jordan). This limits generalizability and should be highlighted more clearly in the discussion.
Some sections in the introduction are overly dense with citations. Consider streamlining for better readability and flow.
Author Response
Reviewers 3
Comments and Suggestions for Authors
This is a methodologically sound mixed study that addresses an important public health issue in a low- and middle-income population. It examines how maternal education, socioeconomic status, and health literacy affect knowledge of risk factors for childhood hearing loss among Jordanian mothers. The study contributes empirical data and theoretical insights, particularly using the Health Belief Model and the Theory of Planned Behavior.
Minor points:
Initial letters of Table and Figure should be capitalized.
Response:
We have capitalized the initial letters of Table and Figure throughout the manuscript, as per the reviewer’s suggestion.
Figure 2 should be mentioned in the main texts.
Response:
We have ensured that Figure 2 is now referenced appropriately in the main text.
Occasional minor inconsistencies in terminology (e.g., “hearing impairment” vs. “hearing loss”) could be standardized.
Response:
We have standardized the terminology throughout the manuscript to consistently use "hearing loss" and replaced "hearing impairment" wherever necessary to avoid inconsistency.
While the thematic analysis is valuable, the qualitative sample is small (n=10), and further participant quotes or broader focus groups could enrich findings.
Response:
We have acknowledged the small qualitative sample size (n=10) and emphasized that larger sample sizes or the use of focus groups in future studies would provide more comprehensive insights.
Acknowledged in limitations, but could be further emphasized. Consideration of observational or clinical data in future studies would be beneficial.
Response:
Response:
We have emphasized in the limitations section that future studies should incorporate observational and clinical data to enhance the validity and depth of findings. This would provide more comprehensive insights into maternal knowledge and early detection of hearing loss.
The sample may not be nationally representative (limited to one region in Jordan). This limits generalizability and should be highlighted more clearly in the discussion.
Response:
We have clarified the limited regional scope of the sample and highlighted that future research should aim for a nationally representative sample to improve the generalizability of the findings.
Some sections in the introduction are overly dense with citations. Consider streamlining for better readability and flow.
Response:
We have streamlined the Introduction section to reduce the density of citations and enhance the overall readability and flow of the text.
Round 2
Reviewer 1 Report
Comments and Suggestions for Authors
The authors have significantly improved their work. Just some minor revisions: I reccomend to spell out AI as Artificial Intelligence at the first appearance at line 688, the proper reference is available (https://www.mdpi.com/1424-8220/24/22/7126; https://pubmed.ncbi.nlm.nih.gov/39598904).
The manuscript states “a mixed-methods design” but does not specify it was embedded with a quantitative priority. Explicitly state in both the abstract and section 2.1 that this is an “embedded mixed-methods study with a dominant quantitative strand supported by qualitative data. This could be stated as well in the title.
The purpose and integration of qualitative findings are not always well-distinguished. Clarify whether qualitative findings were used to explain, validate, or expand upon quantitative results (e.g., quote integration strategies such as "explanatory sequential" or "triangulation"). The qualitative sample size (n = 10) is small and justified only with "data saturation" as a general statement. Briefly explain how saturation was determined (e.g., “No new themes emerged after x interviews”).
The discussion sometimes repeats long segments from results (e.g."A significant positive relationship between SES and maternal knowledge (β = 0.243, t = 43.004, p = 0.000) was confirmed using the results, which led to the acceptance of H2."
This sentence is nearly identical to what was already presented in the Results section and does not add interpretive value. “Mothers could correctly identify factors such as infections and noise exposure, but others had misconceptions or lacked knowledge.” This restates content from the thematic table without further interpretation. Sharpen the focus by synthesizing key findings and linking them directly to existing literature, emphasizing novel contributions.
Author Response
The authors have significantly improved their work. Just some minor revisions: I reccomend to spell out AI as Artificial Intelligence at the first appearance at line 688, the proper reference is available (https://www.mdpi.com/1424-8220/24/22/7126; https://pubmed.ncbi.nlm.nih.gov/39598904).
Response: Thank you for the suggestion. We have updated the manuscript to spell out AI as Artificial Intelligence. Additionally, we have included the proper references for Artificial Intelligence from the provided sources.
The manuscript states “a mixed-methods design” but does not specify it was embedded with a quantitative priority. Explicitly state in both the abstract and section 2.1 that this is an “embedded mixed-methods study with a dominant quantitative strand supported by qualitative data. This could be stated as well in the title.
Response: Thank you for your valuable suggestion. We have revised the manuscript to explicitly state that this is an "embedded mixed-methods study with a dominant quantitative strand". We have incorporated this clarification in the abstract, section 2.1, and title to clearly convey the quantitative priority of the study, supported by qualitative data.
The purpose and integration of qualitative findings are not always well-distinguished. Clarify whether qualitative findings were used to explain, validate, or expand upon quantitative results (e.g., quote integration strategies such as "explanatory sequential" or "triangulation"). The qualitative sample size (n = 10) is small and justified only with "data saturation" as a general statement. Briefly explain how saturation was determined (e.g., “No new themes emerged after x interviews”).
Response: Thank you for your feedback. We have clarified that the qualitative findings were used to expand upon and validate the quantitative results, utilizing an "explanatory sequential" integration strategy. Additionally, we have explained that data saturation was determined when no new themes emerged after 10 interviews.
The discussion sometimes repeats long segments from results (e.g."A significant positive relationship between SES and maternal knowledge (β = 0.243, t = 43.004, p = 0.000) was confirmed using the results, which led to the acceptance of H2."
This sentence is nearly identical to what was already presented in the Results section and does not add interpretive value. “Mothers could correctly identify factors such as infections and noise exposure, but others had misconceptions or lacked knowledge.” This restates content from the thematic table without further interpretation. Sharpen the focus by synthesizing key findings and linking them directly to existing literature, emphasizing novel contributions.
Response: Thank you for your valuable feedback. We have revised the discussion to avoid repetition of the results and have synthesized key findings in a more interpretive manner. We have also linked the results to existing literature and emphasized novel contributions, particularly the role of socioeconomic status (SES) in shaping maternal knowledge, and the need for targeted interventions for lower SES groups. We believe these changes enhance the clarity and depth of the discussion.